# Large Lobular Capillary Hemangioma Associated with Ingrown Toenail: Histopathological Features and Case Report

Antonio Córdoba-Fernández [1,*] , María Dolores Jiménez-Cristino [1] and Victoria Eugenia Córdoba-Jiménez [2]

1 Departamento de Podología, Universidad de Sevilla, C/Avicena s/n, 41009 Sevilla, Spain; mjimenez45@us.es
2 Private Practice, C/Dr. Fleming 13, Bajo B, Castilleja de la Cuesta, 41950 Sevilla, Spain; luna.s__16@hotmail.com
* Correspondence: acordoba@us.es

**Abstract:** Lobular capillary hemangioma (LCH-PG) is a type of pyogenic granuloma characterized by proliferating blood vessels that resemble conventional granulation tissue. Granulation tissue is very often seen in association with ingrown toenails. Despite the close relationship between both entities, LCH-PG shows clinically different behaviors, such as rapid growth and frequent recurrence. Currently, it is unknown exactly how the different etiological factors contribute to the formation of differences between entities. We present a case of a large LCH-PG associated with chronic onychocryptosis in a 26-year-old man. Histopathological features included extensive signs of ulceration, hyperkeratosis, and patchy epidermal acanthosis with the presence of fibrous septa with lobular areas beneath the ulcerative area. The presence of stroma with a marked proliferation of blood vessels with wall thickening and mixed-type inflammatory changes was also characteristic. In advanced stages of onychocryptosis, as presented here, conventional granulation tissue or pyogenic granuloma can be clinically difficult to distinguish from other benign or malignant neoplasms. Histological examination is mandatory, and excisional biopsy can provide a definitive diagnosis.

**Keywords:** capillary hemangioma; piogenic granuloma; granulation tissue; ingrown toenail; onychocryptosis

## 1. Introduction

Lobular capillary hemangioma (LCH) is a histological type of pyogenic granuloma (PG) characterized by proliferating blood vessels organized in lobular aggregates, although superficially the lesion frequently resembles conventional granulation tissue (GT) [1]. Currently, LCH and PG are considered synonyms: the term LCH is typically used by pathologists, and PG is typically used by dermatologists.

Periungual tissues are often subject to acute or chronic mechanical trauma and are susceptible to the development of proliferative lesions. A common example is ingrown toenail or onychocryptosis, which in advanced stages is characterized by the overgrowth of GT arising from the lateral nail fold as a consequence of repetitive mechanical trauma from an offending nail border or nail plate [2]. The role of trauma in the induction of nail tumors is well known, and pathology is mandatory in these cases to exclude a PG [3]. Although GT and PG are frequent benign lesions in the nail unit, the presence of GT or PG may clinically masquerade malignant tumors, such as amelanotic melanoma, squamous or basal cell carcinoma, and Kaposi's sarcoma [4–7].

Usually, in advanced stages of the ingrown toenail, biopsy shows epithelial growing with accompanying GT; this histological pattern can be more or less complex and can mimic from PG to well-differentiated squamous cell carcinoma [8]. Although a biopsy can provide a definitive diagnosis, most patients with GT associated with onychocryptosis are not biopsied or are biopsied inappropriately [9]. We present a case of large LCH-PG secondary to an advanced chronic ingrown toenail with unusually rapid growth. Despite

the close relationship between PG and conventional GT, PG shows clinically different behavior, such as rapid growth and frequent recurrence. However, since malignant lesions such as amelanotic melanoma and sarcoma can masquerade as PG, early diagnosis and treatment can be crucial.

## 2. Case Report

A 26-year-old soccer player came to the clinic with a history of an ingrown toenail of more than a year on the great toe of his left foot. On examination, a slightly painful, exudative, exophytic and polypoid periungual neoplasm covering the nail plate was observed (Figure 1a). The patient reported that he had noticed that the lesion grew rapidly in a few weeks until it completely covered the nail surface. He had been treated in our clinic for a history of previous onychocryptosis in the hallux of the right foot nine years ago. An anteroposterior radiograph of the left foot was taken to rule out osteitis or osteomielitis. Periosteal changes with the presence of osteophytes were observed (Figure 1b). Surgical excisional biopsy was performed under local anesthesia and antibiotic prophylaxis was provided, with 2 g of cephalexin taken orally 30 min before placing the digital tourniquet (Figure 1c). Electrocoagulation was used to prevent bleeding from the surgical site after removal of the tourniquet and local application with nitrofural ointment and a bandage was performed.

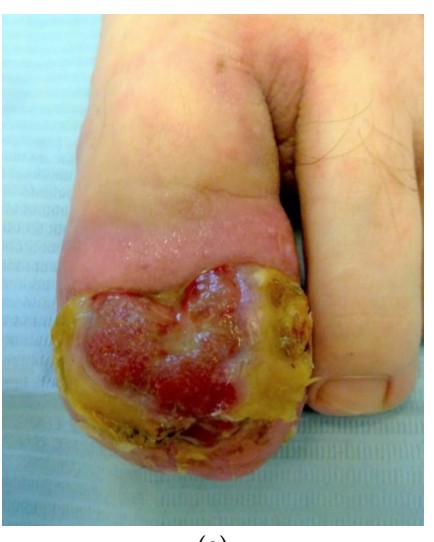
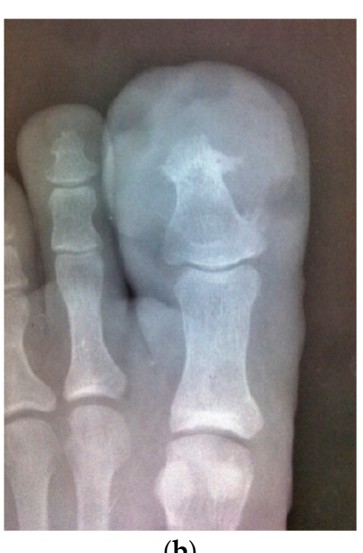
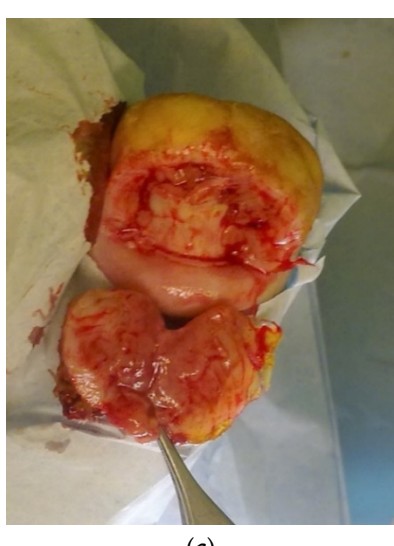

(**a**)                                        (**b**)                                        (**c**)

**Figure 1.** Ulcerated cutaneous lesion covering the nail plate (**a**). Radiographic image shows periosteal changes with the presence of osteophytes in the distal phalanx (**b**). Excisional biopsy shows polypoidaspect of lesion (**c**).

The excisional biopsy revealed a gross irregular fragment of tissue measuring $35 \times 20 \times 5$mm in which an irregular lesion with an ulcerated appearance was identified on the surface. Microscopic examination revealed a tumor with lobular areas located beneath the ulcerative area covered by intact epithelium, hyperkeratosis, and patchy epidermal acanthosis. A variable number of acute and chronic infiltrate of inflammatory cells within the stroma of the superficial lobules could be observed (Figure 2a). The central area consisted of marked proliferation of blood vessels with wall thickening and tiny lumen which werelined by flattened endothelial cells (Figure 2b). Foci of abundant collagen were observed in the stroma, giving a fibrous appearance (Figure 2c).

Based on histological findings, the diagnosis of LCH-PG was established.Once thediagnosis was confirmed, a wedge excision was performed, together with partial nail ablation and matrixectomy of both nail borders, sanitization of the devitalized tissues, and osteophytes resection (Figure 3a).One and a half years after the surgical scission, the lesion resolved completely and has not recurred (Figure 3b).

**Figure 2.** Histopathological panoramic features of lobular capillary hemangioma at low magnification. Fibrous septa and collarettes of the adnexal epithelium that partially surround the lesion (**a**). Lobular areas located beneath the ulcerative area covered with a layer of fibrin and entrapped neutrophils (**a**,**b**). Marked proliferation of blood vessels with wall thickening lined by flattened endothelial cells can be observed (**c**). Stroma with an area of abundant collagen fibers and mixed infiltrate of inflammatory cells around the central blood vessel can be observed (**d**).

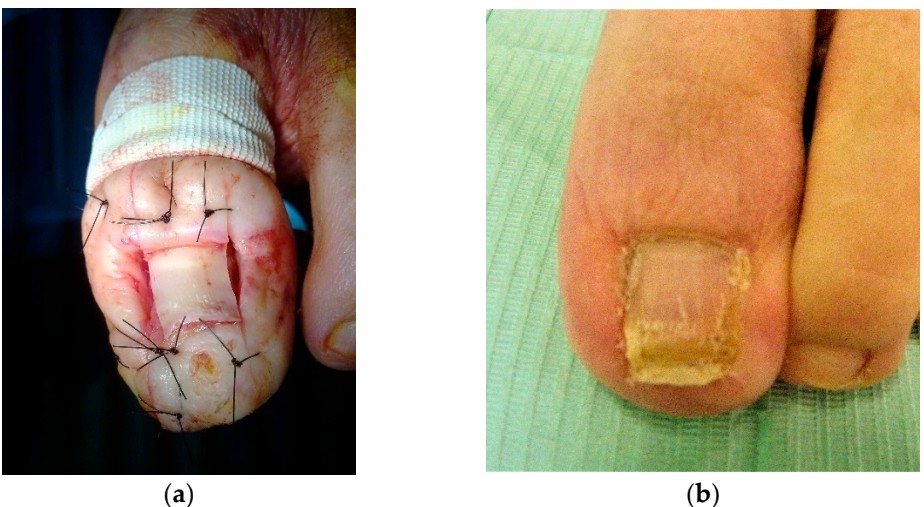

**Figure 3.** Immediate postoperative appearance (**a**). Aspect of hallux toe at year and a half (**b**).

### 3. Discussion

Toenails are often the subject of acute or chronic mechanical trauma due to footwear and sports activities. Advanced stages of onychocryptosis are characterized by GT overgrowth arising from the lateral nail fold as a consequence of repetitive mechanical trauma from the nail spur. In chronic ingrown toenail, hypergranulation tissue with or without fibrosis can occur following chronic soft tissue inflammation. Trauma can contribute to initial GT and can trigger and facilitate proliferation or histological transformation of large tissue, which can generally exceed normal tissue margins. Several reports have described different benign and malign transformations from secondary GT to chronic onychocryptosis [10,11].

PGs are benign vascular neoplasms of unknown etiology. Nail PGs are common, as solitary bleeding nodules or as multiple lesions of several digits. Although nail PGs have been associated with several causes, such as systemic drug treatment, peripheral nerve injury, or inflammatory systemic diseases, mechanical trauma is the most common cause of PG associated with onychocryptosis, retronychia, or onycholysis, especially in the great toe [3].

LCH-PG is a histological type of PG characterized by a lesion that is not a granulomatous lesion at all, even if it may show in its superficial ulceration some resemblance to the GT of a healing wound. Due to the lobulated vascular pattern exhibited by it in its fully developed stage, Mills et al. proposed to rename it LCH; however, this term as a substitute for PG has not yet been universally accepted [12]. LCH-PG presents specific pathological and immune-histochemical characteristics, characterized by the proliferation of newly formed small vessels, oedema, mixed cell infiltrate, and fibrous septa and collarettes of the adnexal epithelium that partially surround the lesion. As in the case presented here, this distinctive pattern becomes more evident when the epidermis regenerates and when a fibrosis surrounds the capillary tufts [1,13,14]. In the present case, immuno-histochemistry was not performed since the endothelial cells could be observed and a definitive diagnosis could be made with only *Masson's trichrome* staining. While local infection is currently ruled out as an etiologic factor for PG, we considered that although not specific enough to allow for a definitive identification of species, when fungal or nonfungal organism infections are suspected, GMS stain can be useful for differential diagnosis [15].

The pathogenesis of LCH-PG pathogenesis is not fully understood and should not be confused with GT. Genetic studies suggested that PG is the result of tissue injury, followed by an altered wound healing response, during which vascular growth is driven by gene mutations [16]. Thus, this favors the idea that PG is a reactive lesion resulting from tissue injury and not a true tumor. LCH-PG shows distinct biological behavior compared to conventional GT, such as rapid growth, multiple occurrence, and frequent recurrence. During the wound healing process, capillary endothelial cell and myofibroblast apoptosis play an important role in the regression from GT to scar tissue, and some studies indicate that PG shows a lower rate of apoptosis and a more frequent expression of regulatory apoptosis proteins than GT [17]. We believe that the LCH-PG in our patient was caused by the transformation of the initial GT toenail tissue as a consequence of sustained chronic inflammation.

Piraccini et al. have classified periungual PG according to pathogenesis, described clinical and pathological characteristics, and provided guidelines for the correct diagnosis and treatment according to etiological factors. The authors recommend surgery and/or curettage in cases of retronychia and onychocryptosis with foreign body penetration [3]. In severe cases, such as the one introduced here, surgery was especially indicated due to the need to remove periungual hyperplasic tissue and the osteophytes present as a consequence of reactive periostitis. Furthermore, wedge scission together with partial matrixectomy of both nail borders turned out to be an effective etiological treatment to prevent recurrence after two years of follow-up.

## 4. Conclusions

A solitary large periungual LCH-PG, as in the present case, is uncommon, and it can grow rapidly from a few millimeters to several centimeters within a few weeks' time. Complete excision is the first-choice therapy. We consider that in cases like this, it is of paramount importance to check for endothelial atypia, hyperchromasia, and mitoses to rule out hemangioendothelioma or low-grade angiosarcoma. Some malignant lesions may show similar clinical characteristics; therefore, histopathological analysis is critical for a definitive diagnosis.

**Author Contributions:** A.C.-F., M.D.J.-C. and V.E.C.-J. took care of the patient. A.C.-F. wrote the manuscript. All authors have read and agreed to the published version of the manuscript.

**Funding:** This research received no external funding.

**Institutional Review Board Statement:** The study was conducted in accordance with the Declaration of Helsinki. The paper is exempt from ethics committee approval as only one case was reported.

**Informed Consent Statement:** The patient provided written informed consent to publish the case, including the publication of images.

**Data Availability Statement:** The patient's data are not publicly available on legal and/or ethical grounds.

**Acknowledgments:** We thank Manuel Salguero Villadiego for his assistance in performing the histopathological analysis and for imaging and consultation related to histopathology.

**Conflicts of Interest:** The authors declare no conflict of interest.

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
