# Peer review of "Large Lobular Capillary Hemangioma Associated with Ingrown Toenail: Histopathological Features and Case Report"

_dermatopathology, doi:10.3390/dermatopathology9030031_

Round 1

Reviewer 1 Report

Please note: LCH and PG are synonyms: LCH used by the pathologists mostly; PG used by the dermatologists mostly. Two names - one entity (LCH being favored in newer textbooks).

You might add one sentence or two in order to clarify your concept: a "non-mushroom-like configurated PG" that looks like granulation tissue but is a site-specific LCH/PG.

Please point out that in cases like this one it is of paramount importance to check for endothelial atypia/hyperchromasia/mitoses in order to rule out variants of hemangioendothelioma / low-grade angiosarcoma.

Author Response

Dear reviewer, I really appreciate your feedback.

  1. In introduction section, we have introduced the following paragraph " 

    "Lobular capillary hemangioma (LCH) is a histological type of pyogenic granuloma (PG) characterized by proliferating blood vessels organized in lobular aggregates, although superficially the lesion frequently resembles conventional granulation tissue (GT)[1]. Currently, LCH and PG are considered synonyms, the term LCH is typically used by pathologists and PG by dermatologists."

  2. In Conclusions section, we have introduced the following sentence: "We consider that in cases like this it is of paramount importance to check for endothelial atypia, hyperchromasia, and mitoses to rule out hemangio-endothelioma or low-grade angiosarcoma.                                            Thank you very much

Reviewer 2 Report

The aim of the paper is to raise awareness of LCH-PG lesions of the toe; highlight that it can mimic malignant tumors, and urge treatment to avoid complications. 

  • Overall the article is interesting, well written and the image quality is excellent.  
  • LCH often regresses on it's own, why do you think this didn't? 
  • Was there a GMS stain performed? Do you think this would have been helpful? 
  • What were other differential diagnoses you were considering? 

Author Response

Dear reviewer, thank you for your comments

In discussion section, we have included the following paragraphs to answer your questions: 

  1. "Although local infection is currently ruled out as an etiologic factor for PG, we considered that although not specific enough to allow for a definitive identification of species, when fungal or nonfungal organism infections are suspected, GMS stain can be useful for differential diagnosis[16]". A new reference has been included.
  2. "Some studies indicate that PG shows a lower rate of apoptosis and a more frequent expression of regulatory apoptosis proteins than GT [18]. We believe that LCH-PG in our patient was caused by transformation of the initial GT toenail tissue as a consequence of sustained chronic inflammation."
  3.  In conclusion section we have included the following sentence: "We consider that in cases like this it is of paramount importance to check for endothelial atypia, hyperchromasia, and mitoses to rule out hemangio-endothelioma or low-grade angiosarcoma".                                          Thank you very much